# A New Human Platelet Lysate for Mesenchymal Stem Cell Production Compliant with Good Manufacturing Practice Conditions Preserves the Chemical Characteristics and Biological Activity of Lyo-Secretome Isolated by Ultrafiltration

**DOI:** 10.3390/ijms23084318

**Published:** 2022-04-13

**Authors:** Katia Mareschi, Alessia Giovanna Santa Banche Niclot, Elena Marini, Elia Bari, Luciana Labanca, Graziella Lucania, Ivana Ferrero, Sara Perteghella, Maria Luisa Torre, Franca Fagioli

**Affiliations:** 1Department of Public Health and Paediatrics, The University of Turin, Piazza Polonia 94, 10126 Torino, Italy; alessiagiovannasanta.bancheniclot@unito.it (A.G.S.B.N.); elena.marini@edu.unito.it (E.M.); franca.fagioli@unito.it (F.F.); 2Stem Cell Transplantation and Cellular Therapy Laboratory, Paediatric Onco-Haematology Division, Regina Margherita Children’s Hospital, City of Health and Science of Turin, 10126 Torino, Italy; iferrero@cittadellasalute.to.it; 3Department of Pharmaceutical Sciences, University of Piemonte Orientale, Largo Guido Donegani 2/3, 28100 Novara, Italy; elia.bari@uniupo.it; 4Blood Component Production and Validation Center, City of Health and Science of Turin, S. Anna Hospital, 10126 Turin, Italy; llabanca@cittadellasalute.to.it (L.L.); glucania@cittadellasalute.to.it (G.L.); 5Department of Drug Sciences, University of Pavia, Viale Taramelli 12, 27100 Pavia, Italy; sara.perteghella@unipv.it (S.P.); marina.torre@unipv.it (M.L.T.)

**Keywords:** mesenchymal stromal cells, human platelet lysate, extracellular vesicles, lyo-secretome, Good Manufacturing Practice

## Abstract

Recently, we proposed a Good Manufacturing Practice (GMP)-compliant production process for freeze-dried mesenchymal stem cell (MSC)-secretome (lyo-secretome): after serum starvation, the cell supernatant was collected, and the secretome was concentrated by ultrafiltration and freeze-dried, obtaining a standardized ready-to-use and stable powder. In this work, we modified the type of human platelet lysate (HPL) used as an MSC culture supplement during the lyo-secretome production process: the aim was to verify whether this change had an impact on product quality and also whether this new procedure could be validated according to GMP, proving the process robustness. MSCs were cultured with two HPLs: the standard previously validated one (HPL-E) and the new one (HPL-S). From the same pool of platelets, two batches of HPL were obtained: HPL-E (by repeated freezing and thawing cycles) and HPL-S (by adding Ca-gluconate to form a clot and its subsequent mechanical wringing). Bone marrow MSCs from three donors were separately cultured with the two HPLs until the third passage and then employed to produce lyo-secretome. The following indicators were selected to evaluate the process performance: (i) the lyo-secretome quantitative composition (in lipids and proteins), (ii) the EVs size distribution, and (iii) anti-elastase and (iv) immunomodulant activity as potency tests. The new HPL supplementation for MSCs culture induced only a few minimal changes in protein/lipid content and EVs size distribution; despite this, it did not significantly influence biological activity. The donor intrinsic MSCs variability in secretome secretion instead strongly affected the quality of the finished product and could be mitigated by concentrating the final product to reach a determined protein (and lipid) concentration. In conclusion, the modification of the type of HPL in the MSCs culture during lyo-secretome production induces only minimal changes in the composition but not in the potency, and therefore, the new procedure can be validated according to GMP.

## 1. Introduction

Recent studies demonstrate that mesenchymal stromal cells (MSCs) efficacy does not lie in their differentiation potential but in secreting bioactive factors [1,2,3]. MSC-secretome comprises a heterogeneous pool of soluble molecules, including cytokines, chemokines and growth factors, and insoluble nano-microstructured vesicles, known as extracellular vesicles (EVs) [4,5]. The secretome is involved in intercellular communication and is assumed to reproduce stem cells’ therapeutic effects, playing an essential role in regulating numerous physiological processes, including immunomodulatory capacities [3,6,7,8]. Indeed, MSC-secretome is rich in anti-inflammatory cytokines, playing an important role in immune responses. However, the mechanisms by which immunomodulation is exercised are still under investigation: in vitro MSCs have been shown to inhibit T-cell proliferation, thus preventing cytotoxic T-cell development, and to inhibit B-cell proliferation and differentiation [9,10].

MSC-secretome could replace MSCs in therapy with even better safety (cell-free therapy), functionality, and storage. Therefore, for the application of MSC-secretome as a biological product, its manufacturing process for clinical trials should comply with Good Manufacturing Practice (GMP) and be subject to rigorous controls and requirements. In Italy, that production should be manufactured in controlled and accredited structures by the Italian Medicines Agency (AIFA). Moreover, to minimize the risks for patients, the regulatory guidelines provide the limiting of additional components and manipulations [11]. In this regard, we validated the MSC production under GMP conditions using human platelet lysate (HPL) inactivated with psoralen or riboflavin as a substitute for fetal bovine serum (FBS) [12]. Furthermore, we recently introduced an innovative method for HPL production: the standard production method consists of freezing and thawing cycles of the pool of donor platelets and the addition of heparin (HPL-E); in contrast, the new method treats the platelet pool with Ca-gluconate to form a gel clot that is mechanically squeezed to release the growth factors without adding heparin (HPL-S). We demonstrated that the HPL produced with the new method (HPL-S) is safer than the standard one (HPL-E) and that it is GMP compliant. Indeed, HPL-S required fewer manipulations of the culture medium because it is more limpid and without debris; thus, numerous filtrations are not necessary. Moreover, HPL-S did not require the addition of heparin, an animal additive. For these reasons, we validated the use of the HPL-S instead of HPL-E for MSC expansion under GMP conditions [13].

Our study aimed to evaluate the effect of changing such a process parameter (different HPL supplementation) on some selected lyo-secretome properties, such as the quantitative composition of lipids and proteins and the size distribution and membrane marker of the EVs. Then the effect of HPL type was investigated for lyo-secretome anti elastase and immunomodulatory activities.

## 2. Results and Discussion

This work investigated the effect of BM-MSC supplementation with HPL-E or HPL-S on their secretome’s physico-chemical and biological properties. First, the Bari et al. production process method was performed to purify MSC-secretome from cell culture supernatants and freeze-dry it, obtaining a ready-on-the-shelf and stable powder [9,14,15]. This process is summarized in Figure 1. In detail, three MSC cell lines were isolated from three bone marrow (BM) samples and expanded simultaneously with HPL-E and HPL-S in the culture medium, as recently described by Mareschi et al. [13]. The characteristics of the BM-MSC samples are shown in Table 1.

It may be supposed that for secretome, similar to any other biologic medicinal product derived from living cells, a unique relationship exists between the production process and the characteristics of the resultant final product [16]. This relationship can be summarized by the phrase “The Product is the process” [17]. Any minimal change during the preparation process—e.g., the supplement used during culture conditions—can strongly affect MSC-secretome qualitative and quantitative composition, with unpredictable and possibly profound consequences on its final characteristics and biological effect(s) [14,18]. Therefore, since we validated the use of HPL-S for the expansion of MSCs replacing the HPL-E [13], secretomes obtained after supplementation of MSCs with HPL-S and HPL-E must be characterized and compared in biological activity. First, some essential properties, such as the quantitative composition of lipids and proteins and the size distribution of the particles (EVs), were evaluated. We selected these tests as they are inexpensive and straightforward, and they can easily and quickly provide performance indicators of the production process and define a primary pharmaceutical quality parameter. Figure 2 reports the total protein and lipid content in the freeze-dried products. The different cell lines used for batch production strongly influenced the amount of proteins and lipids per mg of the final product (*p* < 0.05). In detail, cell line 3 produced significantly higher amounts of proteins and lipids than cell lines 1 and 2 (Figure 2). Treatment with HPL-E or HPL-S did not significantly change the amount of lipids or proteins in the freeze-dried products obtained from cell lines 1 and 2 (*p* > 0.05). Conversely, for cell line 3, higher protein and lipid amounts were obtained after supplementation with HPL-E (*p* < 0.05).

Minimal changes in EV particle size and size distribution were revealed depending on the cell line considered and the supplementation (HPL-E or HPL-S) used. Regarding the mean diameter, the supplementation with HPL-E induced the release of bigger EVs (diameter mean of 198.17 ± 10.45 nm of LSD) than the supplementation with HPL-S (diameter mean of 186.1 ± 10.225 nm of LSD) (*p* < 0.05), while the cell line had no significant effect except for HPL-S between cell line 1 and 2. Furthermore, the interaction between HPL supplementation and cell lines showed significant differences (*p* < 0.001), indicating that the supplementation with HPL-E or HPL-S has an unpredictable effect on the mean size of EVs: for cell lines 1 and 3, bigger EVs were obtained from MSCs cultivated with HPL-E. In comparison, for cell line 2, bigger EVs were released from the MSCs cultivated with HPL-S. Furthermore, the mode was significantly influenced by HPL supplementation and the cell line considered (*p* < 0.05): for cell lines 2 and 3, the mode of the EVs released was bigger than cell line 1 (respectively, *p* < 0.001 and *p* < 0.005), especially when treated with HPL-E. Only in cell line 1 was there a significant difference between the mode of the EVs released that was bigger when supplemented with HPL-E (*p* < 0.05). Regarding the particle size distribution, HPL-E increased the d_10_ of the EVs released (d_10_ mean of 107.98 ± 4.27 nm) compared with HPL-S (d_10_ mean of 102.61 ± 4.14 nm). Statistical differences were also observed between cell lines (Figure 3).

To complete the physico-chemical analysis, freeze-dried secretomes were analyzed for their immunophenotypic profile for surface markers CD90, CD9, CD29, CD73, CD63, CD44, CD146, CD81, and CD105 (Figure 4), as suggested by the International Society of Extracellular Vesicles [19]. Flow cytometry results show no significant differences between the surface markers of EVs obtained from BM cell lines expanded with HPL-E or HPL-S.

The biological activity of secretomes obtained from the two supplementations was also investigated. At first, the ability of HPL-S and HPL-E freeze-dried secretomes to inhibit the elastase enzyme was evaluated (Figure 5). All samples showed good anti-elastase properties; of note, at the highest concentration considered (20 mg/mL), the anti-elastase activity for cell lines 2 and 3 was higher than in the positive control. Furthermore, the secretome obtained from cell line 3 showed a significantly higher anti-elastase activity with respect to secretomes produced by cell lines 1 and 2 (*p* < 0.05). Similar batch-to-batch variability, linked to the different cell lines used for their production, was revealed for non-human species, such as horses [15]. Previous work has indicated that the anti-elastase activity is related to alpha-1-antitrypsin (AAT) in MSC-secretome, the main elastase inhibitor in the lung, and other proteins involved in protease/antiprotease balance [6]. Accordingly, the higher activity for secretomes obtained from cell line 3 is likely attributable to the higher amount of proteins and lipids produced by this cell line (see Figure 2). Likely, secretomes from cell line 2, with higher protein content with respect to the ones from cell line 1, are more active, and a dose-dependent percentage increase in anti-elastase inhibition was revealed by increasing the concentration/mg of freeze-dried secretome, and thus the total protein and AAT content for all the samples. Overall, the supplementation with HPL-E or HPL-S did not significantly modify the anti-elastase activity (*p* = 0.5685).

The enzymatic kinetics of anti-elastase activity were then elaborated with a Michaelis–Menten kinetics model to extrapolate the Vmax and Km values, as reported in Table 2. By analyzing Vmax and Km values, it is possible to predict the inhibition mechanism of the sample. In detail, the positive control was shown to act on elastase as a mixed inhibitor, as it significantly increased Km and reduced Vmax values compared with the negative control (*p* < 0.05). For all the freeze-dried secretomes produced by the three different cell lines, the Km was increased (especially at the highest doses). In contrast, Vmax values were not significantly different from the negative control ones. This suggests that the lyo-secretome from differently cultured MSCs inhibited the elastase enzyme in the same way, according to our previous results [6].

The immunomodulatory properties of secretomes obtained from cell lines 1, 2, and 3 with the supplementation of HPL-S and HPL-E were evaluated. In particular, PBMCs were isolated from three peripheral blood samples (Table 3), activated with phytohemagglutinin (PHA) for 72 h, and treated with a dose of lyo-secretome equivalent/PBMC ratio 1:10. This secretome dosage was identified to obtain the same immunomodulatory activity as MSCs, and it was determined based on previous work that performed dose/response tests for the soluble fraction, the EVs fraction, and the whole secretome [9]. The analyses performed were: cytokine production, cell generations during the culture of PBMCs, and finally, a flow cytometry assay on lymphocyte subpopulations.

The ELISA assay detected the following cytokines: IL-4, IL-6, IL-10, Il-12, and TNF-α (Figure 6). The concentration of all the cytokines considered was increased after PHA treatment, thus confirming the model’s suitability. Freeze-dried secretomes modulated cytokine production in a dose-dependent manner. The immunomodulatory properties were not significantly modified by the HPL supplement (*p* > 0.05) but from the cell line used to produce secretomes. Specifically, the lyo-secretomes obtained from cell line 3 were more effective in reducing IL-4 and IL-6 produced by activated PBMCs, while the secretomes from cell lines 1 and 2 were ineffective. Similarly, IL-12 was significantly reduced after treatment with secretomes prepared from cell lines 2 and 3 but not from lyo-secretomes prepared from cell line 1. All samples significantly increased the production of anti-inflammatory IL-10 and reduced the production of pro-inflammatory TNF-α. Secretomes obtained from cell line 3 were the most effective in reducing TNF-α concentration and the least effective in increasing IL-10 concentration. Again, the higher activity of secretomes obtained from cell line 3 is likely attributable to the higher amount of proteins and lipids produced by this cell line (see Figure 2).

Thanks to the flow cytometry assay, we analyzed lymphocyte subpopulations and the ability of the secretome to modulate them when activated PBMCs of three donors were treated with different batches of lyo-secretome. Again, the immunomodulatory properties were not significantly modified by the HPL supplement (*p* > 0.05) but by the cell line used to produce secretomes (Figure 7): we observed significant differences that depended on the sample of MSC used for the production (see Table 1). In detail, the total number of lymphocytes and CD3+ cells increased significantly after PHA treatment and were then decreased by treating with secretomes obtained from cell lines 1 and 2. Conversely, CD4+ and CD8+ cell numbers remained unmodified after PHA and lyo-secretome treatment.

These data are in accordance with some reports in the literature [10,20] describing that the secretome has an action on the secretion of cytokines by lymphocytes but that it has no proliferative effect on lymphocytes. Therefore, the non-significance is probably due to a lower immunomodulating effect of the lyo-secretome compared with the use of MSCs in co-culture immunomodulant experiments, as previously studied and demonstrated by Mareschi et al. [21].

Overall, these results demonstrate that it is not the supplementation with HPL-S or HPL-E but the BM-MSCs (1, 2, and 3) that modify the secretome’s chemical composition and biological activities. Indeed, the cell line used to prepare lyo-secretomes were shown to be the primary source of variability: secretomes obtained from cell line 3 were highly abundant in proteins and lipids and consequently showed higher anti-elastase and immunomodulatory properties. Such variability in quali-quantitative composition and biological activity results from the production process, which standardizes the final product in terms of cell equivalents: a precise quantity of the final product is obtained with a specific number of cells without considering how many proteins and lipids were produced. Therefore, this work confirms the need for a different standardization of the final product that, as recently suggested by Mocchi and colleagues [15,22], could be in terms of specific protein (and lipid) concentration instead of cell equivalent concentration. In other words, during production, the final product must be concentrated to reach a specific protein (and lipid) concentration rather than an equivalent cell concentration.

In addition, we have previously demonstrated that HPL-S could be considered a better cell culture supplement for MSC expansion under GMP conditions instead of the standard HPL-E. Indeed, during the production, the addition of animal additives such as heparin and the medium filtration steps are avoided. This contributes to mitigating the risk of contamination during cell production and makes the HPL-S production process more GMP compliant [13]. In this regard, no significant differences between the secretome obtained from MSCs cultivated with HPL-E and HPL-S have been shown, indicating a possible substitution of the standard HPL-E with the new, safer and more GMP-compliant HPL-S. Consequently, the lyo-secretome derived from MSCs cultivated with HPL-S is more compliant with the required GMP standards.

## 3. Materials and Methods

### 3.1. Materials

Acetone was bought from Carlo Erba reagents, Milan, Italy. Bovine serum albumin, epigallocatechin gallate (EGCG) mannitol, N-succinyl-tri-l-alanine-4-nitroanilide, Nile Red, phosphatidylcholine, porcine pancreatic elastase (PPE) were purchased from Sigma-Aldrich, Milan, Italy.

### 3.2. HPL Preparation

The HPL was manufactured by the Blood Component Production and Validation Center, City of Science and Health of Turin, S. Anna Hospital (C.P.V.H), from two buffy coat-derived platelet concentrates (BC-PCs), as recently described [13].

Briefly, HPL-E was produced using a standard method consisting of freezing and thawing cycles of the pool of donor platelets (PLTs) and the addition of heparin. At the same time, HPL-S treated the PLTs pool with Ca-gluconate to form a gel clot that is mechanically squeezed without adding heparin. For this work, three batches of HPL-E/S were produced and tested.

### 3.3. Isolation and Culture of Human MSCs

The study adhered to the Declaration of Helsinki: human BM samples were collected after submission of written consent, in accordance with the “City of Health and Science Hospital of Turin, Pediatric Onco Hematology, Regina Margherita Children’s Hospital” Ethics Committee. MSCs were harvested in unfiltered BM collection waste bags (Baxter Healthcare. Deerfield, IL, USA), usually used for transplants. The bag was washed with Dulbecco’s phosphate-buffered saline (PBS, Thermo Fisher Scientific, Massachusetts, USA), and the cells were collected and washed at 200× *g* for 5 min. An aliquot of whole BM was counted and plated directly in Cell Factory Systems (Thermo Fisher Scientific) at 1 × 10^4^ cell/cm^2^ in six-well multiwell flasks at 1 × 10^4^ cell/cm^2^ and 1 × 10^5^ cell/cm^2^ to evaluate colony-forming unit fibroblasts (CFU-F). Three lines were expanded: BM-01, BM-02, and BM-03. The culture medium used was Alpha Minimum Essential Medium Eagle (α-MEM, Sigma-Aldrich, MERCK) supplemented with 1% L-glutamine (Sigma-Aldrich, MERCK), 1% penicillin/streptomycin (Sigma-Aldrich, Merck KGaA, Darmstadt, Germania), and 10% of HPL-E or HPL-S (Mareschi et al. 2021). The culture was maintained at 37 °C in a 5% CO_2_ atmosphere. Non-adherent cells were removed after 7–10 days, and the adherent cells were re-fed every 3–4 days. When the cells reached about 80% of confluence, they were detached with trypsin/EDTA (Sigma-Aldrich, MERCK) for 5 min at 37 °C, expanding them until the third passage. The cells at each passage were frozen in α-MEM with 5% albumin and 10% dimethyl sulfoxide (DMSO).

As suggested by the International Society for Cell and Gene Therapy for defining multipotent mesenchymal stromal cells [23] and by scientists participating in the GISM (Gruppo Italiano Staminali Mesenchimali) Working Group [24], we performed immunophenotype analysis and assessed differential and proliferative potential. The flow cytometry assay was performed using the following monoclonal antibodies (mAb): anti-CD90 fluorescein isothiocyanate (FITC), CD73 phycoerythrin (PE), CD34-CD14-CD45 (FITC), CD105 allophycocyanin (APC), CD19-CD146 (APC), and HLA-DR (PE). Specifically, 1 × 10^6^ cells were stained with 10 μL mAb conjugated with fluorochrome for 20 min at 4 °C in the dark. The labeled cells were washed with PBS (200× *g* for 10 min) and analyzed using Beckman Coulter NAVIOS (Krefeld, Germany) with the Navios program (Vs. 1.2, Beckman Coulter, Krefeld, Germany). The percentage of positive cells was calculated using the un-stained cells as a negative control. Mean fluorescence intensity (MFI) was analyzed on the positive cells.

To analyze their multipotent capacity, the MSCs were cultured in), adipogenic, and chondrogenic media (Miltenyi Biotec, Bergisch Gladbach, Germany) for three weeks, according to the manufacturer’s instruction. Osteogenic differentiation was demonstrated by the accumulation of crystalline hydroxyapatite on Von Kossa staining, adipogenic differentiation with intracellular lipid vesicles assessed with oil red O, and chondrogenic differentiation by alcian blue staining. The CFU-F was evaluated in a colony established one week later in acetone–methanol and colored with the May Grunwald Dye. The cells were counted at an optical microscope using Burker Chamber. The growth of the cells was calculated with the population doubling (PD), and their expansion was expressed in terms of cumulative PD (cPD). Their viability was assessed by staining with Trypan Blue Method (Sigma-Aldrich, St. Louis, MO, USA).

### 3.4. Lyo-Secretome Production and Analysis

#### 3.4.1. Secretome Concentration, Purification, and Lyophilization

After being collected, culture media were centrifuged at 3500× *g* for 10 min to eliminate cell debris and apoptotic bodies. Then, according to previously published procedures [6,22,25,26], the recovered supernatants were ultrafiltered using the tangential flow filtration system KrosFlo^®^ Research 2i, from Spectrum Laboratories, Milan, Italy. A filtration module with molecular weight cut-off (MWCO) of 5 kDa (Spectrum Laboratories, Milan, Italy) with a superficial area of 790 cm^2^ was chosen to retain both free soluble factors and EVs produced using AD-MSCs. At first, the supernatants were concentrated at 0.5 × 10^6^ cell equivalents per mL and then diafiltered using sterilized ultrapure water. Throughout all the ultrafiltration steps, the feed stream shear rate was maintained between 2000 and 6000 s^−1^, and the transmembrane pressure did not exceed 30 psi. Mannitol (Sigma-Aldrich, Milan, Italy) was finally dissolved at 0.5% *w*/*v*, and the resulting solution was frozen at −80 °C and freeze-dried (Christ Epsilon 2–16D LSCplus) at 8 × 10^−1^ mbar and −50 °C for 72 h. The obtained lyo-secretome was stored at −20 °C until use (maximum 12 months). Next, the cell equivalents were calculated, dividing the total cell number used for production and the obtained milligrams of lyo-secretome: each mg of lyo-secretome corresponds to 0.1 × 10^6^ cell equivalents.

#### 3.4.2. Lyo-Secretome Analysis

Freeze-dried samples were characterized as reported below.

##### Total Protein and Lipid Content

Proteins were dosed using the BCA Protein Assay Kit from Thermo Fisher Scientific (Milan, Italy). The working reagent solution was prepared according to the manufacturer’s instructions, added to each sample (or standard) in a 1:1 ratio, and incubated at 37 °C for 2 h. A microplate reader measured the absorbance at 562 nm (Synergy HT, BioTek, Potton, UK). A calibration curve was prepared using bovine serum albumin in the concentration range of 0–100 μg/mL; R^2^ was 0.99.

Lipids were dosed using Nile Red assay [6,15]. Nile Red was solubilized in acetone (3.14 M) and then diluted 100× *g* in PBS. Nile Red solution was added to each sample (or standard) in a 1:9 ratio and incubated at room temperature for 5 min. Synergy HT measured fluorescence at fixed wavelengths (530/25 excitation and 645/40 emission). A calibration curve was prepared using phosphatidylcholine in the concentration range of 0–20 μg/mL; R^2^ was 0.99.

##### EV Particle Size Distribution

Nanoparticle tracking analysis (NTA) was used to determine the EVs’ particle size and concentration in all free-dried samples. The NanoSight NS 300 equipment (Malvern Instruments) was used. The freeze-dried powder was first dispersed in deionized water (1 mg/mL), gently shackled, and analyzed. All measurements were repeated three times and were carried out at room temperature with a detection angle of 90°.

##### EV Flow Cytometry

The EV particle characterization was performed through cytofluorometric analysis using the following monoclonal antibodies: anti-CD90, CD9, CD29, CD73, CD63, CD44, CD146, CD81, and CD105. Therefore, 50 μL of lyo-secretome was stained with 2 μL of mAb conjugated with a fluorochrome (for CD81, 10 μL was used) for 20 min at 4 °C in the dark. The labeled EVs were washed with PBS and analyzed as previously described [13].

##### Growth Factor Analysis

An aliquot of each HPL was collected to evaluate concentration levels of IL4, IL6, IL10, IL12, and TNF-α. The analysis was performed using an ELISA kit (Invitrogen, ThermoFisher Scientific, Waltham, Massachusetts, Stati Uniti) following the manufacturer’s instructions. Briefly, 50–100 μL of standards and samples were plated in duplicate on the well already coated with the antibody and incubated at room temperature (RT) for 2–3 h. Four washes were performed, and then 50–100 μL of biotinylated antibody 1X for 1 h at RT was added and incubated for 1 h at RT. Another four washes were performed before adding streptavidin-horseradish peroxidase. At the end of the procedure, we washed them four times and incubated them with an appropriate Stabilized Chromogen, conferring at the reaction a coloring with an intensity (evaluated with a spectrophotometer at a wavelength of 450 nm) directly proportional to the concentration of the factor. Data are expressed as the concentration in picograms per milliliter.

#### 3.4.3. Biological Activity

##### In Vitro Anti-Elastase Activity

Freeze-dried powders were dispersed in deionized water at final concentrations of 1, 2, 10, and 20 mg/mL. Phosphate buffer at pH 6.8 was added to the enzyme PPE, reaching a 0.5 units/mL concentration. TRIS-HCl buffer pH 8 was added to the substrate N-succinyl-tri-l-alanine-4-nitroanilide, reaching a final concentration of 0.41 mM. An amount of 12.5 μL of the sample was incubated with 12.5 μL of elastase solution, 50 μL of the substrate solution, 12.5 μL of PBS, and 50 μL of TRIS-HCl. The reaction kinetics were evaluated at 410 nm wavelength with a UV–vis microplate reader (Synergy HT, Milan, Italy). The assay was carried out in triplicate. A reaction mixture without samples was used as a negative control. EGCG was used as positive control and tested at the final concentration of 0.6 mg/mL. Anti-elastase activity percentage was calculated using the following formula: Anti-elastase activity % = (A − B)/A × 100, where A is the absorbance of the negative control, and B is the absorbance of the tested solution.

##### Immunomodulatory Activity

PBMCs were isolated from peripheral blood (PB) units from voluntary donations of healthy subjects to CPVH, with prior informed consent. Three PB samples (A, B, C, see Table 3) and three lots of lyo-secretome were used for each HPL: HPL-E 181, 185, 202; HPL-S 182, 186, 203 8—See Table 1). The PBMCs were separated from buffy coats by centrifugation on a Ficoll Histopaque-1077 density gradient (Sigma-Aldrich St. Louis, USA), according to the manufacturer’s instructions. To isolate macrophage-monocyte discriminating lymphocyte cells, PBMCs were counted and plated for 1 h in T75 bottles with RPMI 1640 medium (Sigma-Aldrich St. Louis, MO, USA) and 10% FBS at 37 °C in a 5% CO_2_ atmosphere. After 1 h, non-adherent cells were resuspended in medium and plated in 6-well multiwell bottles at 3 × 10^6^ cells/cm^2^ for 72 h. The conditions tested were: (i) unstimulated PBMCs; (ii) PBMCs stimulated with PHA; (iii) PBMCs stimulated with PHA + lyo-secretome-HPL-E (#1,#3,#5); (iv) PBMC stimulated with PHA + lyo-secretome-HPL-S (#2,#4,#6).

Following the procedure’s instructions, cells were stained with CFSE dye (Sigma-Aldrich, St. Louis, MO, USA) prior to seeding to assess lymphocyte generations. The 25 mg of HPL-E/S-lyo-secretome was reconstituted with 2.5 mL of PBS, yielding 1 × 10^6^ cell equivalents. The lyo-secretome equivalent/PBMC ratio was 1:10. To activate T lymphocytes, PBMCs were stimulated with PHA (2.5 μg/mL). At seeding and at the end of the immunomodulation experiment, the PBMCs were characterized through multiparametric flow cytometric analysis, which allowed the identification of the following subsets of T lymphocytes with the following antibodies: CD45 + (Krome Orange), CD3 + (PC 5.5), CD4 + (APC), CD8+ (PE). Briefly, 2 × 10^5^ cells were stained with 10 μL fluorochrome-conjugated mAb (20 min at 4 °C, in the dark), and the labeled PBMCs were washed and analyzed using Navios software (Vs. 1.2, Beckman Coulter, Krefeld, Germany). Non-labeled antibody cells were used as a negative control.

### 3.5. Statistical Analysis

Raw data were processed using STATGRAPHICS XVII (Statpoint Technologies, Inc., Warrenton, VA, USA). A linear generalized analysis of variance model (ANOVA) was used, followed by Fisher’s least significant difference (LSD) procedure to estimate the differences between the means. HPL supplement and cell lines were considered fixed factors for each analysis, while protein and lipid content, EV mean diameter, mode, d_10_ and d_90_, and the cytokine concentration (for the immunomodulatory experiments) were considered the response variable. The enzymatic kinetics of anti-elastase activity were elaborated with a model of Michaelis–Menten kinetics y = (Vmax × x)/(Km + x), where y is the absorbance at time x, Km is the moment in which the activity is equal to half the maximum, and Vmax is the maximum speed of the enzyme [27]. The curve parameters were calculated, and all graphics were performed with the GraphPad Prism software version 8 for the Windows^®^ platform. For each curve, Vmax and Km were analyzed with an ANOVA model. Finally, the differences between the groups were analyzed with the LSD test for multiple comparisons. Statistical significance was fixed at *p* < 0.05. Moreover, we used in the graphics the symbols #, ##, ###, and #### to indicate a significant difference between the means of the different batches and the symbols *, **, ***, **** to indicate a significant difference between the means of 2 HPLs, respectively, with a *p* value of <0.05, *p* < 0.01, *p* < 0.005, and *p* < 0.0001.

## 4. Conclusions

The new human platelet lysate supplementation for MSCs culture induces only a few changes in the physico-chemical properties selected as indicators, such as protein and lipid content and EVs size distribution. On the other hand, this effect is not associated with differences in the biological activity of lyo-secretome in terms of inhibition of the elastase enzyme and immunomodulatory activity. Instead, the most important source of batch variability derives from the cell line: in other words, the intrinsic MSCs properties have a very significant effect (i) on the properties of the secretome selected as process quality control and (ii) on biological activity defined for potency tests (anti-elastase and immunomodulant activity). The intrinsic variability in MSCs to produce lyo-secretome affects the quality of the finished product and can be mitigated by concentrating the final product to reach a determined protein (and lipid) concentration rather than an equivalent cell concentration.

In conclusion, modifying a critical process parameter, such as the type of HPL in the culture of MSCs, induces minimal and measurable changes in the quality of the lyo-secretome. Therefore, the new procedure can be validated according to GMP.

## Figures and Tables

**Figure 1 ijms-23-04318-f001:**
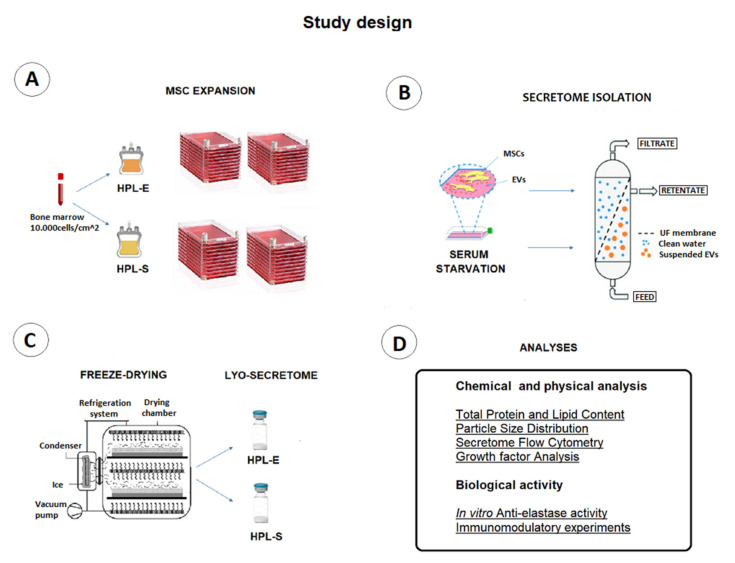
Study design summary. Alpha-MEM was added with HPL-E or HPL-S for MSC cell cultures (**A**). The culture supernatants were collected, ultrafiltered (**B**), and freeze-dried (**C**). The obtained lyo-secretome was analyzed for physico-chemical characteristics and used to perform immunomodulation experiments (**D**) to test the lyo-secretome biological activity.

**Figure 2 ijms-23-04318-f002:**
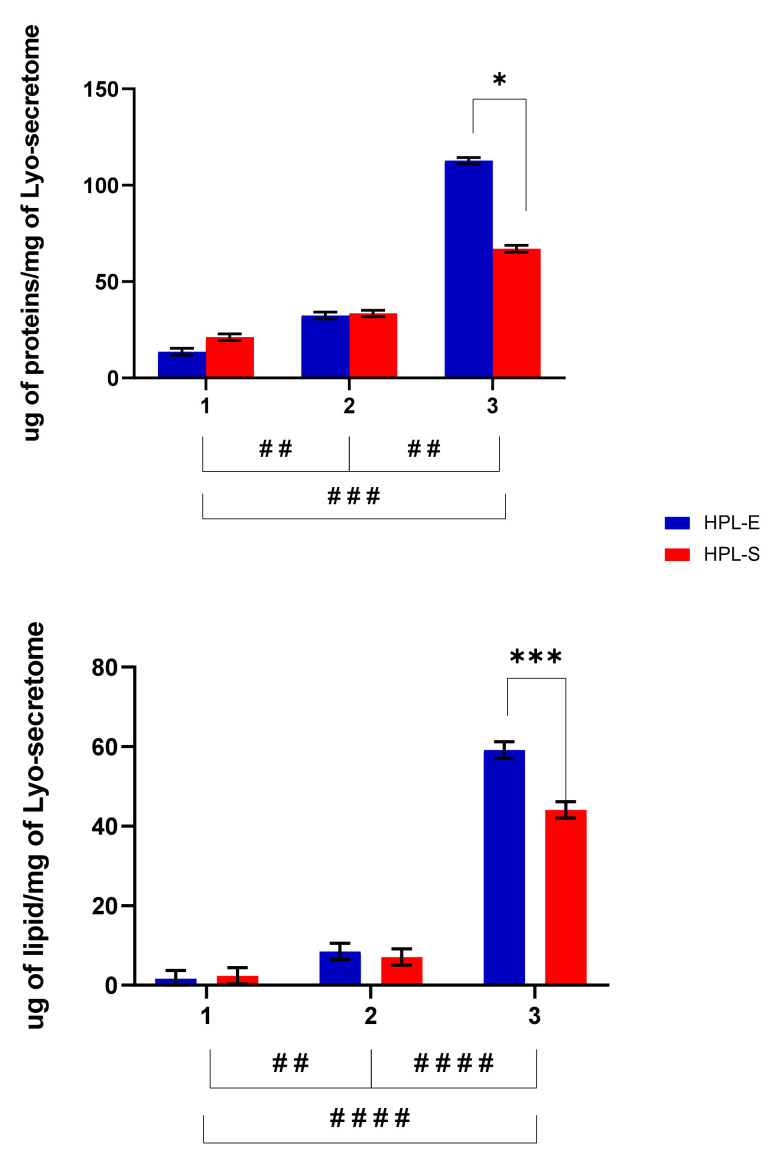
Protein and lipid content of freeze-dried secretome obtained from 3 different BM cell lines (1, 2, and 3) expanded with HPL-E or HPL-S. Multifactor ANOVA, mean values ± LSD, n = 3 are indicated. The symbols ##, ### and #### indicate a significant difference between the means of the different batches (respectively, *p* < 0.01, *p* < 0.005 and *p* < 0.0001), while the symbols * and *** indicate a significant difference between the two HPLs in the same batch (respectively, with *p* < 0.05 and *p* < 0.005).

**Figure 3 ijms-23-04318-f003:**
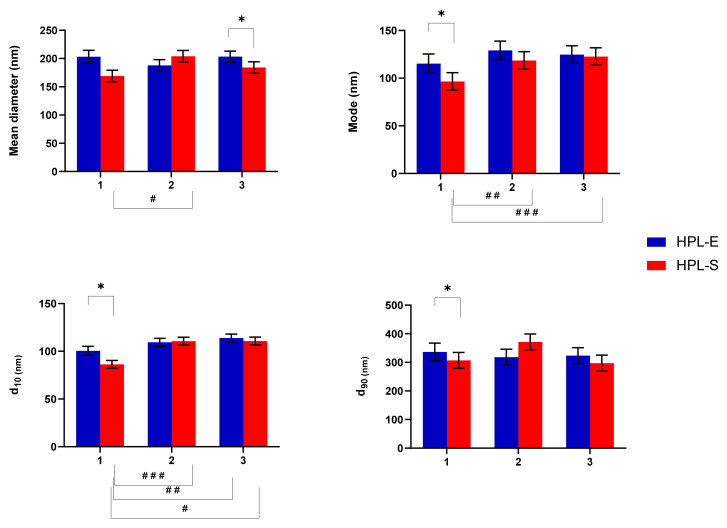
Particle size and size distribution of freeze-dried EVs obtained from BM cell lines 1, 2, and 3 expanded in HPL-S or HPL-E. Multifactor ANOVA, mean values ± LSD, *n* = 3 are indicated. The symbols #, ##, and ### indicate a significant difference between the means of the different batches (respectively, *p* < 0.05, *p* < 0.01, and *p* < 0.005), while the symbols * indicate a significant difference between the two HPLs in the same batch (*p* < 0.05).

**Figure 4 ijms-23-04318-f004:**
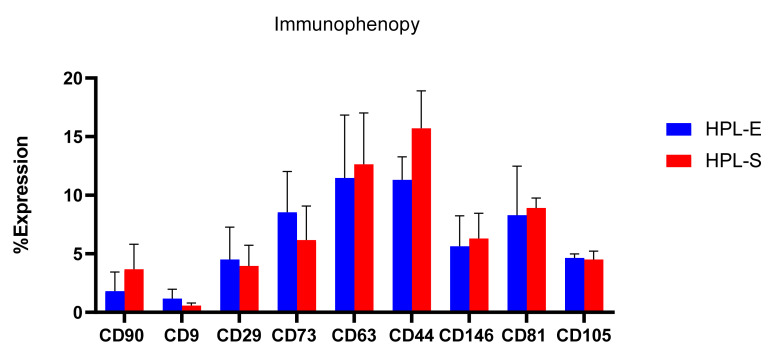
Flow cytometry analysis of EVs obtained from BM cell lines 1, 2, and 3 expanded in HPL-S or HPL-E. Multifactor ANOVA, mean values ± LSD, *n* = 3 are illustrated. No statistical differences are reported between two HPLs in all analyzed markers.

**Figure 5 ijms-23-04318-f005:**
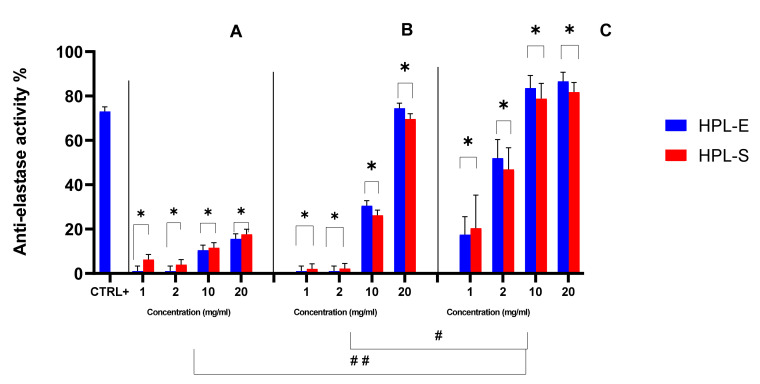
Anti-elastase activity percentage of freeze-dried MSC-secretome obtained from BM cell lines 1 (**A**), 2 (**B**), and 3 (**C**) expanded in HPL-S or HPL-E. CTR +: EGCG at 0.6 mg/mL in deionized water. Multifactor ANOVA, mean values ± LSD, *n* = 3 are reported. * indicates a significant difference between the means (*p* < 0.05) between HPL-E and HPL-S, while # and ## indicate a significant difference between the means of the experiments (respectively, with a *p* < 0.05 and *p* < 0.01).

**Figure 6 ijms-23-04318-f006:**
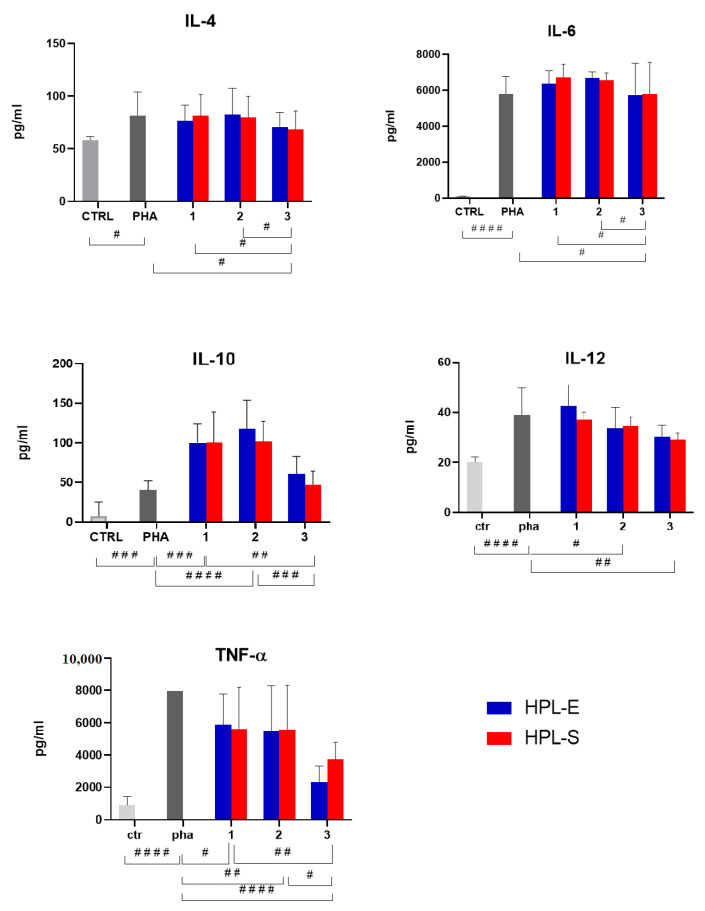
Modulation of cytokine production from PHA-activated PBMCs after treatment with freeze-dried MSC-secretome obtained from cell lines 1, 2, and 3 and expanded with LyoS-HPL-E or LyoS-HPL-S. Multifactor ANOVA, mean values ± LSD, *n* = 3 are reported. The symbols #, ##, ### and #### indicate a significant difference between the means of the different batches (respectively, *p* < 0.05, *p* < 0.01, *p* < 0.005 and *p* < 0.0001). No significant differences are reported between 2 HPLs in all cytokine analyses.

**Figure 7 ijms-23-04318-f007:**
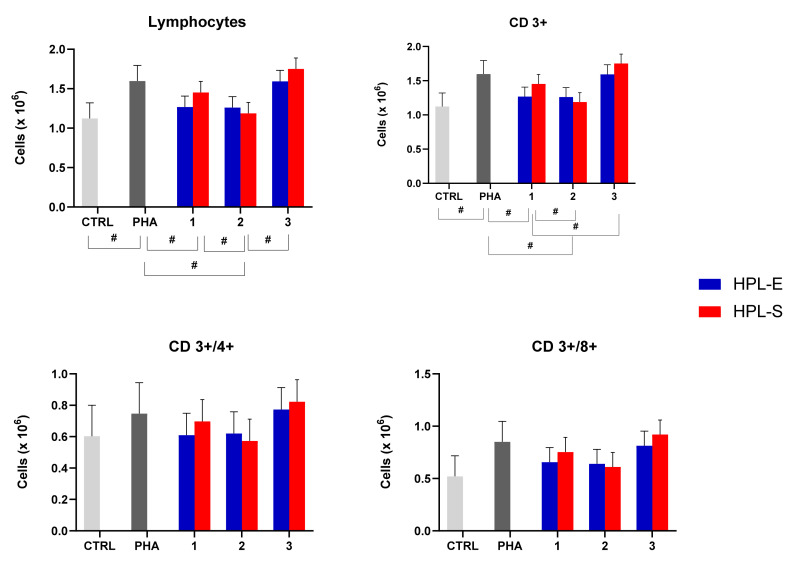
Analysis of the number of lymphocyte subpopulations from PHA-activated PBMCs after treatment with lyo-secretome from cell lines 1, 2, and 3 expanded with HPL-E or HPL-S. Multifactorial ANOVA, mean values ± LSD, n = 3 are illustrated. The symbol # indicate a significant difference between the means of the different batches (respectively, *p* < 0.05). No significant differences are reported between 2 HPLs in all analyzed cell-subsets.

**Table 1 ijms-23-04318-t001:** Bone marrow samples collected with information on relative donors’ age and sex/gender, the ID of mesenchymal stromal cells cell lines (BM-MSC-02 04, 05) derived from BM samples and the ID of lyo-secretome isolated by BM-MSCs cultivated in HPL-E (LyoS-HPL-E #1, #3,#5) and in HPL-S (LyoS-HPL-S #2, #4,#6).

*ID Bone Marrow*	*Age*	*Sex*	*ID Batch*	*ID Lyo-Secretome*
**BM-MSC-02**	23	Male	1	LyoS-HPL-E-#1 LyoS-HPL-S-#2
**BM-MSC-04**	22	Male	2	LyoS-HPL-E-#3 LyoS-HPL-S-#4
**BM-MSC-05**	40	Male	3	LyoS-HPL-E-#5 LyoS-HPL-S-#6

**Table 2 ijms-23-04318-t002:** Vmax and Km values for each sample were analyzed. Multifactor ANOVA, mean values ± LSD, *n* = 3.

Sample	Cell Line	HPL	Concentration (mg/mL)	Km ± SE (95% Confidence Bounds)	Vmax ± SE (95% Confidence Bounds)
MSC-secretome	1	HPL-E	1	10.76 ± 0.4623 (9.855, 11.75)	0.9039 ± 0.01406 (0.8762, 0.9336)
			2	10.16 ± 0.4419 (9.295, 11.11)	0.8901 ± 0.01362 (0.8632, 0.9190)
			10	11.11 ± 0.4695 (10.19, 12.11)	0.8993 ± 0.013986 (0.8719, 0.9288)
			20	14.52 ± 0.539 (13.47, 15.66)	0.9076 ± 0.0141 (0.8800, 0.9373)
		HPL-S	1	13.79 ± 0.3604 (13.09, 14.54)	0.9746 ± 0.0104 (0.9543, 0.9960)
			2	12.99 ± 0.5581 (11.91, 14.17)	0.9551 ± 0.01636 (0.9234, 0.9895)
			10	16.41 ± 0.7882 (14.92, 18.08)	0.9794 ± 0.02082 (0.9397, 1.023)
			20	20.5 ± 0.907 (18.77, 22.43)	1.024 ± 0.02214 (0.9818, 1.071)
	2	HPL-E	1	23.33 ± 0.4493 (22.45, 24.25)	1.015 ± 0.007955 (0.9996, 1.031)
			2	28.2 ± 1.076 (26.13, 30.45)	1.04 ± 0.01764 (1.006, 1.077)
			10	63.96 ± 3.411 (57.60, 71.37)	1.088 ± 0.03579 (1.021, 1.165)
			20	72.55 ± 3.58 (44.22, 138.0)	0.4393 ± 0.07628 (0.3274, 0.6907)
		HPL-S	1	34.74 ± 1.51 (31.89, 37.91)	1.121 ± 0.02375 (1.076, 1.171)
			2	34.44 ± 0.8741 (32.75, 36.26)	1.101 ± 0.01357 (1.075, 1.129)
			10	57.69 ± 1.143 (55.47, 60.05)	1.127 ± 0.01329 (1.101, 1.154)
			20	128.4 ± 32.76 (82.35, 238.3)	0.9585 ± 0.1841 (0.6982, 1.573)
	3	HPL-E	1	34.13 ± 0.3613 (32.17, 36.24)	0.9749 ± 0.00454 (0.9477, 1.004)
			2	68.53 ± 0.8625 (44.17, 118.1)	0.8694 ± 0.02455 (0.6718, 1.262)
			10	283.4 ± 12.3523 (210.0, 420.4)	0.7623 ± 0.03406 (0.5911, 1.081)
			20	470.2 ± 52.4223 (347.0, 711.1)	1.509 ± 0.05425 (1.147, 2.215)
		HPL-S	1	43.84 ± 2.2539 (39.29, 49.08)	1.109 ± 0.02478 (1.047, 1.180)
			2	43.17 ± 10.2547 (19.75, 121.5)	0.7328 ± 0.05458 (0.5115, 1.415)
			10	166. 6 ± 6.2659 (138.1, 206.7)	0.6446 ± 0.02563 (0.5567, 0.7676)
			20	267.9 ± 8.4452 (241.3, 300.2)	1.162 ± 0.02425 (1.063, 1.282)
CTR−				19.51 ± 0.3846 (18.46, 20.62)	0.8908 ± 0.01204 (0.8729, 0.9095)
CTR+				74.28 ± 0.6643 (64.65, 86.14)	0.5644 ± 0.03426 (0.5169, 0.6226)

**Table 3 ijms-23-04318-t003:** PBMC samples enrolled with relative age and sex for secretome biological study.

*ID PBMC*	*Age*	*Sex*
**A**	46	Female
**B**	33	Female
**C**	46	Female

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
