# Peer review of "A New Human Platelet Lysate for Mesenchymal Stem Cell Production Compliant with Good Manufacturing Practice Conditions Preserves the Chemical Characteristics and Biological Activity of Lyo-Secretome Isolated by Ultrafiltration"

_ijms, 2022, doi:10.3390/ijms23084318_

Round 1

Reviewer 1 Report

A new Human Platelet Lysate for Mesenchymal Stem Cell Pro-duction compliant with Good Manufacturing Practice conditions preserves the chemical characteristics and biological ac-tivity of lyo-secretome isolated by ultrafiltration.

The authors wrote an interesting article for the effect of MSC-EV, however I have some comments regarding the characterization of their EV from their samples. The comments are as follows:

Major Comments

  1. Intro: “MSC-secretome comprises a heterogeneous pool of soluble molecules, including cytokines, chemokines and growth factors and insoluble nano-microstructured vesicles, known as extracellular vesicles (EVs).”

Add References to end of sentence as missing references:

  • Shephard AP, Yeung V, Clayton A, Webber JP. Prostate cancer exosomes as modulators of the tumor microenvironment. J Cancer Metastasis Treat 2017;3:288-301. http://dx.doi.org/10.20517/2394-4722.2017.32
  • Yeung V, Willis GR, Taglauer E, Mitsialis SA, Kourembanas S. Paving the Road for Mesenchymal Stem Cell-Derived Exosome Therapy in Bronchopulmonary Dysplasia and Pulmonary Hypertension. Stem Cell-Based Therapy for Lung Disease. 2019 Aug 7:131–52. doi: 10.1007/978-3-030-29403-8_8. PMCID: PMC7122497.

  1. Intro: “The secretome is involved in intercellular communication and is assumed to reproduce the therapeutic effects of stem cells themselves, playing an essential role in regulating numerous organs and physiological processes, including immunomodulatory capacities[3,4].”

    Add a list of references with [3,4]: - these references also reports the functional capacity of MSC-EV (or their secretome in numerous models) and their immunomodulatory capacity.

  • Nikfarjam, S., Rezaie, J., Zolbanin, N.M. et al. Mesenchymal stem cell derived-exosomes: a modern approach in translational medicine. J Transl Med 18, 449 (2020). https://doi.org/10.1186/s12967-020-02622-3
  • McKay, Tina B et al. “Extracellular Vesicles in the Cornea: Insights from Other Tissues.” Analytical cellular pathology (Amsterdam) vol. 2021 9983900. 22 Jul. 2021, doi:10.1155/2021/9983900

  1. Figure 1. Should be expanded to be largest as it is very hard to read.
  2. The letters is not very clear in the figures when discussing significant differences. There should be an arrow pointing between the two variables you want to compare to.

    I would omit the “same letters” as the reader can see there is no differences as this detracts from the main point you are making from the significant differences between your two groups.

    In Figure 2A – I would keep “d” and “e” and omit the “a” and “c”
    In Figure 2B – I would keep “c” and “d” and omit the “a” and “b”

    If there are significant differences – I would add a * value on the graph and an arrow signifying the differences you want to compare.

  3. The authors have not shown the characterization of their MSC in accordance to the minimum criteria for MSCs as set by The International Society for Cellular Therapy (ISCT) position statement (Dominici et al 2006). 

    The authors have mentioned this in their methods but have not shown this in their Figures. The MSC has not been tested for: MSC must express CD105, CD73 and CD90, and lack expression of CD45, CD34, CD14 or CD11b, CD79alpha or CD19 and HLA-DR surface molecules.
  4. The authors must show MSC can differentiate to osteoblasts, adipocytes and chondroblasts in vitro as per The International Society for Cellular Therapy (ISCT) position statement (Dominici et al 2006).
  5. All Figures you should remove the letters between your comparisons (as Figures 4 -6 did not states the significance of the letters in the Figure legend). To keep it more readily accessible to the readers – place all * values with those differences in significant differences.

    For those with no significant differences (“same letter”) you should omit those letters as the comparison is not shown.

  6. The authors have not showed sufficient evidence of EV characterization (main or supplemental) in accordance with the with the minimum guidelines set by the International Society of Extracellular Vesicles (ISEV) (Thery et al 2018 JEV).

    The authors should also include western blot analysis showing at least 3 minimal markers and one negative control as per MISEV 2018. In additional the authors should show the morphology of the EVs by transmission electron microscopy. We will be able to clarify the differences in morphology for HPL-E vs HPL-S.

  1. Can the authors state how their EV-dose was used for their functional experiments and justification for doing so? How did the authors decide to proceed with this? As the authors characterized their EV preparation with both protein and lipid concentrations.

  2. Can the authors state how significant the differences are within their Figures? *P<0.05, **P<0.01, ***P<0.001? That should be some clarity into the level of significance found. It is very hard to understand what a, b or c is? The authors should incorporate an arrow between two experimental groups
  3. The authors should also include some additional limitations and propose that the soluble secretome (devoid of EVs) was not tested in their functional assays. They soluble secretome could in fact have different effects on their immunomodulatory capacity.

Minor Comments

  1. Abstract: In addition, their anti-elastase activity and “immunomulant” properties were also analyzed. Typo?
  2. Intro: Iin Italy, that production should be manufactered in controlled and accredited structures by the Italian Medicines Agency (AIFA). Typo?
  3. Check the consistency of spacing throughout the text.
    “We validated the MSC production in GMP conditions using Human Platelet Lysate (HPL) inactivated with psoralen or riboflavin as a substitute for foetal bovine serum (FBS )[8].Recently” There is a space missing after [8]
  4. Can the authors use superscript/subscript effectively please.
    Methods:
    After 1 hour, non-adherent cells were resuspended in medium and plated in 6-well Multiwell bottles at 3 × 106 cells/cm2 for 72 hours

    It infers you have 3 x 106 cells… please check for consistency through the journal.

Overall, the article does require revision and reassurances that the isolation method and nomenclature of EV remains consistent in accordance with the ISEV criteria requirements. The study remains interesting but there are some issues that must be addressed.

Author Response

Dear review 1,

please find enclosed the revised manuscript entitled “A new Human Platelet Lysate for Mesenchymal Stem Cell Pro-duction compliant with Good Manufacturing Practice conditions preserves the chemical characteristics and biological activity of lyo-secretome isolated by ultrafiltration.

We wish to thank you for giving us the opportunity to revise and improve our work.

In this revised version we took into consideration the reviewers’ criticisms and modified the text accordingly.

Moreover, whole manuscript has been reviewed by native English speaker.

All alterations in the revised manuscript are tracked using the      "Track Changes" function in Microsoft Word.

We believe that following the reviewers’ suggestions the new version has been much improved, and hope that it is now suitable for the publication in your journal.

Below is our response to the your comments point to point

Yours sincerely

Katia Mareschi

A new Human Platelet Lysate for Mesenchymal Stem Cell Pro-duction compliant with Good Manufacturing Practice conditions preserves the chemical characteristics and biological activity of lyo-secretome isolated by ultrafiltration.

The authors wrote an interesting article for the effect of MSC-EV, however I have some comments regarding the characterization of their EV from their samples. The comments are as follows:

Major Comments

  1. Intro: “MSC-secretome comprises a heterogeneous pool of soluble molecules, including cytokines, chemokines and growth factors and insoluble nano-microstructured vesicles, known as extracellular vesicles (EVs).”

Add References to end of sentence as missing references:

  • Shephard AP, Yeung V, Clayton A, Webber JP. Prostate cancer exosomes as modulators of the tumor microenvironment. J Cancer Metastasis Treat 2017;3:288-301. http://dx.doi.org/10.20517/2394-4722.2017.32
  • Yeung V, Willis GR, Taglauer E, Mitsialis SA, Kourembanas S. Paving the Road for Mesenchymal Stem Cell-Derived Exosome Therapy in Bronchopulmonary Dysplasia and Pulmonary Hypertension. Stem Cell-Based Therapy for Lung Disease. 2019 Aug 7:131–52. doi: 10.1007/978-3-030-29403-8_8. PMCID: PMC7122497.

We have added these references as suggested by the reviewer.

  1. Intro: “The secretome is involved in intercellular communication and is assumed to reproduce the therapeutic effects of stem cells themselves, playing an essential role in regulating numerous organs and physiological processes, including immunomodulatory capacities [3,4].”

    Add a list of references with [3,4]: - these references also reports the functional capacity of MSC-EV (or their secretome in numerous models) and their immunomodulatory capacity.

  • Nikfarjam, S., Rezaie, J., Zolbanin, N.M. et al. Mesenchymal stem cell derived-exosomes: a modern approach in translational medicine. J Transl Med 18, 449 (2020). https://doi.org/10.1186/s12967-020-02622-3
  • McKay, Tina B et al. “Extracellular Vesicles in the Cornea: Insights from Other Tissues.” Analytical cellular pathology (Amsterdam) vol. 2021 9983900. 22 Jul. 2021, doi:10.1155/2021/9983900

We have also added these references as suggested by the reviewer

  1. Figure 1. Should be expanded to be largest as it is very hard to read

We have modified the figure to be largest and easier to read

  1. The letters is not very clear in the figures when discussing significant differences. There should be an arrow pointing between the two variables you want to compare to.

    I would omit the “same letters” as the reader can see there is no differences as this detracts from the main point you are making from the significant differences between your two groups.

    In Figure 2A – I would keep “d” and “e” and omit the “a” and “c”
    In Figure 2B – I would keep “c” and “d” and omit the “a” and “b”

    If there are significant differences – I would add a * value on the graph and an arrow signifying the differences you want to compare.

Because we used      the two-factor or two-way analysis of variance (ANOVA) to compare the mean level of a dependent variable for one independent variable and SIMULTANEOUSLY to compare the mean level for other independent variables (in our case different batches  and different HPL), in all the figures we indicated with # the differences between the batches and with  * the differences between HPL. Moreover, p value < 0.05, < 0.01. <0.005 and <0.001 were indicated respectively with 1 (* or #), 2 (++ or ##), or 3 (*** or ###) or (**** or ####) symbols.

  1. The authors have not shown the characterization of their MSC in accordance to the minimum criteria for MSCs as set by The International Society for Cellular Therapy (ISCT) position statement (Dominici et al 2006).

We have recently published a paper which well describes the characterisation of MSCs used for preparation of the Lyosecretome used in this manuscript.

We added the reference and parts in the Introduction, Material & Methods and in the Discussion

  1. The authors have mentioned this in their methods but have not shown this in their Figures. The MSC has not been tested for: MSC must express CD105, CD73 and CD90, and lack expression of CD45, CD34, CD14 or CD11b, CD79alpha or CD19 and HLA-DR surface molecules.

As explained in the point 4 above, these data are published in the ref 13 which have added in the revised paper

  1. The authors must show MSC can differentiate to osteoblasts, adipocytes and chondroblasts in vitro as per The International Society for Cellular Therapy (ISCT) position statement (Dominici et al 2006).

As explained in the point 4 above, these data are published in the ref 13 which we have added in the revised paper

  1. All Figures you should remove the letters between your comparisons (as Figures 4 -6 did not states the significance of the letters in the Figure legend). To keep it more readily accessible to the readers – place all * values with those differences in significant differences.
    For those with no significant differences (“same letter”) you should omit those letters as the comparison is not shown.

As reported in the point 2, we added the * and # to indicate the statistically significant  p values in all the figures

  1. The authors have not showed sufficient evidence of EV characterization (main or supplemental) in accordance with the with the minimum guidelines set by the International Society of Extracellular Vesicles (ISEV) (Thery et al 2018 JEV).
  2. The authors should also include western blot analysis showing at least 3 minimal markers and one negative control as per MISEV 2018. In additional the authors should show the morphology of the EVs by transmission electron microscopy. We will be able to clarify the differences in morphology for HPL-E vs HPL-S.

We thank the referee for these comments (points 7 &8) because they prove that we were not sufficiently clear in reporting the objective of our work. Our intention was not to do an in-depth analysis of the secretome      and various components of its complex protein matrix and its particulate component (including IVs), but only to evaluate the effect of changing a process parameter (type of supplementation), on some simple essential properties, such as the quantitative composition in lipids and proteins, or the size distribution of the particles (EVs), regardless of its proteomic or lipidomic composition. We have selected these tests as they are simple, inexpensive and which can easily and quickly provide control indicators of the production process. Subsequently we focused on few membrane markers, and, in detail     , not on the commonly used ones (set by the International Society of Extracellular Vesicles, Thery et al 2018 JEV) because they are expressed by all EVs as such and, therefore, are less likely to change. following the process change.

In other words, we focused not on the similarities but on the potential differences between the secretomes. However, the in-depth morphological and proteomic investigation had previously been done on lyosecretome obtained from mesenchymal cells, from adipose tissue and bone marrow, from human, canine and equine cells.

https://doi.org/10.3390/cells7110190

https://doi.org/10.2217/nnm-2018-0240

https://doi.org/10.3390/cells8090965

https://doi.org/10.3390/ph14060553

https://doi.org/10.3390/ani11113271

Following these studies, we confirmed that with a precise robust and validated process it was possible to obtain a product with reproducible properties (https://doi.org/10.3390/pharmaceutics13081129). For the same reason we did not even perform the TEM analysis, since the phospholipid bilayer ultrastructure, in our experience, was less likely to change following a change in a process parameter.

To better clarify the above, the aim at the end of the introduction, the discussion, and the conclusions have been amended (see text with changes highlighted)

  1. Can the authors state how their EV-dose was used for their functional experiments and justification for doing so? How did the authors decide to proceed with this? As the authors characterized their EV preparation with both protein and lipid concentrations.

We thank the referee for this comment. In previous works we performed lyosecretome dose / response tests for the soluble fraction, the EVs and the whole secretoma and a secretome dosage was identified to obtain the same immunomodulatory activity of MSCs (https://doi.org/10.3390/cells7110190. In this paper we cho     se EV Lyosecretome equivalent / PBMC ratio was 1:10 and the dose was      quantified in terms of cell equivalents.

This concept was added in results and discussions (see text with changes highlighted, page 8 before table 3). However, since different cell lines produce secretomes with different quantitative composition (roughly expressed in total proteins and lipids) in our recent work (https://doi.org/10.3390/pharmaceutics13081129) we have standardized the production and the final product is concentrated to reach a specific protein (and lipid) concentration instead of cell equivalent concentration.

  1. Can the authors state how significant the differences are within their Figures? *P<0.05, **P<0.01, ***P<0.001? That should be some clarity into the level of significance found. It is very hard to understand what a, b or c is? The authors should incorporate an arrow between two experimental groups.

In accordance with the reviewer, we modified all the figures indicating the levels of significance.

  1. The authors should also include some additional limitations and propose that the soluble secretome (devoid of EVs) was not tested in their functional assays. They soluble secretome could in fact have different effects on their immunomodulatory capacity.

We apologize to the referee for being unclear. In fact, we have evaluated in all the experiments the activity of the Lyosecretome "in toto" and not the activity of the soluble fraction and the corpuscular fraction separately. We had already done these tests in a previous work (https://doi.org/10.3390/cells7110190). Furthermore, in another work (https://doi.org/10.3390/cells8090965) we have determined the activities of EV and the soluble fractions at different molecular weight: high molecular weight (having MW between 100 and 300 kDa) and low molecular weight (having MW between 5 and 100 kDa).

The text has been amended.

Minor Comments

  1. Abstract: In addition, their anti-elastase activity and “immunomulant” properties were also analyzed. Typo?
  2. Intro: Iin Italy, that production should be manufactered in controlled and accredited structures by the Italian Medicines Agency (AIFA). Typo?
  3. Check the consistency of spacing throughout the text.
    “We validated the MSC production in GMP conditions using Human Platelet Lysate (HPL) inactivated with psoralen or riboflavin as a substitute for foetal bovine serum (FBS )[8].Recently” There is a space missing after [8]
  4. Can the authors use superscript/subscript effectively please.
    Methods:
    After 1 hour, non-adherent cells were resuspended in medium and plated in 6-well Multiwell bottles at 3 × 106 cells/cm2 for 72 hours
    It infers you have 3 x 106 cells… please check for consistency through the journal.

We thank the referee, and we reviewed the whole manuscript and modified all points suggested by the referee as minor comments

Overall, the article does require revision and reassurances that the isolation method and nomenclature of EV remains consistent in accordance with the ISEV criteria requirements. The study remains interesting but there are some issues that must be addressed.

The text has been completely reviewed considering all suggestions of the referee

Reviewer 2 Report

The aim of this study was to compare two kinds of Human Platelet Lysates (HPL), a standard one (HPL-E) and a new one (HPL-S). From the same pool of human platelets, two batches of HPLs were obtained, the HPL-E (obtained by repeated freezing and thawing cycles) and the HPL-S (produced through the addition of Ca-gluconate to form a clot and its subsequent mechanical wringing). They subsequently investigated the biochemical and functional properties of the secretome released by mesenchymal stem cells (MSCs) isolated from Bone Marrow (BM) cultured with the addition of the two different HPLs. The data show that there is no significant difference between the secretomes obtained from MSCs cultivated with HPL-E and HPL-s. The authors conclude that a possible substitution of the standard HPL (HPL-E) with the new, safer and more GMP compliant lysate (HPL-S) is possible.

Overall, the data are well described but represent only a marginal advancement of the paper just published by the same group on the same topic

Int. J. Mol. Sci. 202223(6), 3234; https://doi.org/10.3390/ijms23063234

Also, it is not clear whether GMP conditions have now been reached

Author Response

Dear review 2,

please find enclosed the revised manuscript entitled “A new Human Platelet Lysate for Mesenchymal Stem Cell Pro-duction compliant with Good Manufacturing Practice conditions preserves the chemical characteristics and biological activity of lyo-secretome isolated by ultrafiltration.

We wish to thank you for giving us the opportunity to revise and improve our work.

In this revised version we took into consideration the reviewer 2’ criticisms and modified the text accordingly.

Moreover, whole manuscript has been reviewed by native English speaker.

All alterations in the revised manuscript are tracked using the "Track Changes" function in Microsoft Word.

We believe that following the reviewers’ suggestions the new version has been much improved, and hope that it is now suitable for the publication in your journal.

Below is our response to the your comments point to point

Yours sincerely

Katia Mareschi

Comments and Suggestions for Authors

The aim of this study was to compare two kinds of Human Platelet Lysates (HPL), a standard one (HPL-E) and a new one (HPL-S). From the same pool of human platelets, two batches of HPLs were obtained, the HPL-E (obtained by repeated freezing and thawing cycles) and the HPL-S (produced through the addition of Ca-gluconate to form a clot and its subsequent mechanical wringing). They subsequently investigated the biochemical and functional properties of the secretome released by mesenchymal stem cells (MSCs) isolated from Bone Marrow (BM) cultured with the addition of the two different HPLs. The data show that there is no significant difference between the secretomes obtained from MSCs cultivated with HPL-E and HPL-s. The authors conclude that a possible substitution of the standard HPL (HPL-E) with the new, safer and more GMP compliant lysate (HPL-S) is possible.

Overall, the data are well described but represent only a marginal advancement of the paper just published by the same group on the same topic.         

Int. J. Mol. Sci. 202223(6), 3234; https://doi.org/10.3390/ijms23063234

Yes . it is true, we have recently published a manuscript where a part of the material and methods      is the same. However, in this manuscript we performed more analyses to verify that MSCs isolated with a new HPL preserve all MSC immunophenotypic, and multipotent characteristics as described by The International Society for Cellular Therapy (ISCT) position statement (Dominici et al 2006). Then, we also performed additional experiments focusing our attention on the secretome isolated from MSCs with a standard HPL and the new HPL. We have submitted almost simultaneously the 2nd manuscript      in which  the focus and experimental designs are completely different. Finally, the experiments described in this manuscript have allowed us to validate the isolation and expansion method of BM-MSCs with the HPL because we demonstrated that new HPL preserved also the chemical characteristics and biological activity of lyo-secretome isolated by ultrafiltration. We have modified the ref 9 and specified the differences in the text in the  introduction, material &methods and the discussion

Also, it is not clear whether GMP conditions have now been reached.

This manuscript describes a validation process necessary to perform a change control in MSC manufacturing process, according to GMP conditions

Round 2

Reviewer 1 Report

I am happy for the authors amendments and comments for the manuscript. I am happy to accept this manuscript to be published in the IJMS journal.

Reviewer 2 Report

Overall, the authors show that there are no differences between the secretome obtained from MSCs cultivated with HPL-E or HPL-S. An important conclusion is that substitution of the standard HPL-E with the new, safer and more GMP-compliant HPL-S is possible and makes Lyo-secretomes derived from MSCs cultivated with HPL-S also more compliant with required GMP standards.

The authors responded to most comments made and improved the paper.